# Biological Control of a Root-Knot Nematode *Meloidogyne incognita* Infection of Tomato (*Solanum lycopersicum* L.) by the Oomycete Biocontrol Agent *Pythium oligandrum*

**DOI:** 10.3390/jof10040265

**Published:** 2024-04-02

**Authors:** Yuwei Xue, Weishan Li, Mengnan Li, Ningchen Ru, Siqiao Chen, Min Jiu, Hui Feng, Lihui Wei, Paul Daly, Dongmei Zhou

**Affiliations:** 1College of Food and Bioengineering, Henan University of Science and Technology, Luoyang 471023, China; yweixue0912@163.com (Y.X.); lws@ofdc.org.cn (W.L.); 2Key Lab of Food Quality and Safety of Jiangsu Province—State Key Laboratory Breeding Base, Institute of Plant Protection, Jiangsu Academy of Agricultural Sciences, Nanjing 210014, China; lexiuhu@hebeu.edu.cn (M.L.); runingchen@stmail.ujs.edu.cn (N.R.); 33315220@njau.edu.cn (S.C.); fenghui@jaas.ac.cn (H.F.); weilihui@jaas.ac.cn (L.W.); 3Nanjing Institute of Environmental Sciences, Ministry of Ecology and Environment, Nanjing 210042, China; 4College of Landscape and Ecological Engineering, Hebei University of Engineering, Handan 471023, China; 5School of Environment and Safety Engineering, Jiangsu University, Zhenjiang 212013, China; 6Fungal Genomics Laboratory (FungiG), Nanjing Agricultural University, Nanjing 210095, China

**Keywords:** *Pythium oligandrum*, *Meloidogyne incognita*, biological control, tomato, induced resistance

## Abstract

The biocontrol agent *Pythium oligandrum*, which is a member of the phylum Oomycota, can control diseases caused by a taxonomically wide range of plant pathogens, including fungi, bacteria, and oomycetes. However, whether *P. oligandrum* could control diseases caused by plant root-knot nematodes (RKNs) was unknown. We investigated a recently isolated *P. oligandrum* strain GAQ1, and the *P. oligandrum* strain CBS530.74, for the control of an RKN *Meloidogyne incognita* infection of tomato (*Solanum lycopersicum* L.). Initially, *P. oligandrum* culture filtrates were found to be lethal to *M. incognita* second-stage juveniles (J2s) with up to 84% mortality 24 h after treatment compared to 14% in the control group. Consistent with the lethality to *M. incognita* J2s, tomato roots treated with *P. oligandrum* culture filtrates reduced their attraction of nematodes, and the number of nematodes penetrating the roots was reduced by up to 78%. In a greenhouse pot trial, the *P. oligandrum* GAQ1 inoculation of tomato plants significantly reduced the gall number by 58% in plants infected with *M. incognita*. Notably, the *P. oligandrum* GAQ1 mycelial treatment significantly increased tomato plant height (by 36%), weight (by 27%), and root weight (by 48%). A transcriptome analysis of tomato seedling roots inoculated with the *P. oligandrum* GAQ1 strain identified ~2500 differentially expressed genes. The enriched GO terms and annotations in the up-regulated genes suggested a modulation of the plant hormone-signaling and defense-related pathways in response to *P. oligandrum*. In conclusion, our results support that *P. oligandrum* GAQ1 can serve as a potential biocontrol agent for *M. incognita* control in tomato. Multiple mechanisms appear to contribute to the biocontrol effect, including the direct inhibition of *M. incognita*, the potential priming of tomato plant defenses, and plant growth promotion.

## 1. Introduction

*Pythium oligandrum* is an important biocontrol agent that can parasitize or predate fungi and oomycetes and uses multiple mechanisms of action [1,2,3], but it has not yet been investigated for its control of root-knot nematode (RKN)-caused diseases. Predation or parasitism is a major biocontrol mechanism of *P. oligandrum* [4], which, after coming into contact with a plant pathogen, can quickly penetrate the mycelium of the plant pathogen [5]. *P. oligandrum* can also induce plant resistance, and this induced plant resistance involves increases in the production of pathogenesis-related proteins (PRs) and the modulation of hormone signaling pathways [6]. *P. oligandrum* can also promote plant growth, which involves the production, by *P. oligandrum*, of the auxin compound tryptamine [7]. Recently, there has been a report of a *P. oligandrum* preparation called Ecosin controlling a parasitic disease in dogs and cats caused by the ascarid nematodes *Toxocara canis* and *T. cati* [8]. To our knowledge, however, there are no reports of *P. oligandrum* having nematicidal activity towards plant parasitic nematodes or controlling nematode-caused plant diseases.

Root-knot nematodes (RKNs) are plant-parasitic nematodes with an obligate endoparasitic lifestyle that cause major losses to agricultural production [9]. As a major RKN species, *Meloidogyne incognita* causes major losses in global agricultural production [10]. RKNs have a complex life cycle involving plant–parasitic interactions, but there is a limited mechanistic understanding of this life cycle, as reviewed recently by Rutter et al. [11]. As detailed descriptions can be found elsewhere, such as Trudgill and Blok [12], a brief overview of the life cycle of *Meloidogyne* spp. is described. Following molting, the first-stage juvenile transforms into its second stage, J2, and these J2s can infiltrate the plant root tip. The formed giant cells serve as feeding sites and are where *M. incognita* continues to develop, causing root galls in the process. In general, RKN eggs may survive outside of the plant host for at least one year, contributing to their persistence, but the survival ability of the J2s is far more limited, with a recent study showing that the survivability of *M. incognita* J2s depends on lysosome-mediated lipolysis [13].

Tomato (*Solanum lycopersicum* L.) is a prominent vegetable crop with an annual production of ~200 million tonnes, and there are several major fungal and bacterial diseases of tomato [14]. RKNs cause major losses to the tomato industry, and at present the major treatments are chemical fungicides, but there is a need to develop more environmentally friendly alternatives [15]. There is an increasing demand for new nematicides to replace the more hazardous currently used nematicides, as reviewed recently by Desaeger et al. [16]. There are several previous reports of bacterial and fungal species used for biocontrol with nematicidal activity that can control RKN-caused diseases [17,18]. However, the use of biocontrol agents to control RKNs is far rarer than the use of conventional nematicides, and there are many challenges to increasing the use of biocontrol agents, as reviewed recently by Abd-Elgawad and Askary [19]. Recently, *Trichoderma harzianum* was shown to suppress the RKN infestation of tomato, and there was support for induced resistance being one of the biocontrol mechanisms from the data that showed increased levels of defense-related metabolites and increased expression of defense-related genes [20].

Although there are many studies using bacterial or true fungal (from the kingdom Fungi) biocontrol agents, e.g., Zhang et al. [21] recently reviewed nematode interactions with true fungi, there is still a lack of research using oomycete biocontrol agents to control root-knot nematodes such as *M. incognita*. Previously, the *P. oligandrum* GAQ1 strain was shown to parasitize or predate the oomycete broad-host-range plant pathogen *P. myriotylum* and control ginger soft-rot disease [22]. The objectives of this study were to investigate the antagonistic effects of *P. oligandrum* strains GAQ1 and CB530.74 on *M. incognita*, as well as their biocontrol mechanism through in vitro-based antagonism assays, pot trials, and transcriptome analyses.

## 2. Methods

### 2.1. Maintenance and Routine Culturing of P. oligandrum Strains

The *P. oligandrum* strain GAQ1 (Po_GAQ1) is preserved in the China General Microbiological Culture Collection Center (deposit number CGMCC No.17470). As described previously, the Po_GAQ1 strain was isolated from soil from a field where infected ginger was growing in Laiwu, Shandong Province, China [22]. The *P. oligandrum* strain CBS530.74 strain (Po_CBS530.74) was gifted to us by Prof. Daolong Dou, Nanjing Agricultural University, and was described previously by Kushwaha et al. [23]. The *P. oligandrum* CBS530.74 strain was isolated from soil in the Netherlands in 1969 according to culture collection records. The *P. oligandrum* CBS530.74 and GAQ1 strains can be distinguished genetically using barcoding primers for *CoxI,* as was shown in our previous study [22]. The *P. oligandrum* GAQ1 strain has a distinct chrysanthemal colony appearance on V8 agar, while this appearance is not obvious for the *P. oligandrum* CBS530.74 strain.

For routine plate culturing, agar plugs of the strains Po_GAQ1 and Po_CBS530.74 were inoculated on 10% V8 juice agar and incubated at 25 °C for at least 24 h. For the preparation of the V8 juice medium, one 340 mL can of V8 vegetable juice (The Campbell Soup Company, Camden, NJ, USA) was mixed with 3.5 g CaCO_3_ for 2 min, then centrifuged for 5 min at 2377× *g*. Then, the liquid supernatant was filtered through two layers of medical gauze and diluted 10-fold with double distilled water, and 1.5% *w*/*v* agar was added if required and autoclaved at 121 °C for 15 min. For the preparation of culture filtrates and mycelial pellets, 30 agar plugs of 6 mm diameter (from the edge of the colony) were used to inoculate 50 mL of 10% V8 juice medium and incubated at 90 RPM and 25 °C for 3 d until mycelial pellets were produced. The culture was filtered using triple-folded filter paper.

### 2.2. Nematicidal Effect of P. oligandrum Culture Filtrates to M. incognita

From cut-up infected tomato root tissues, the eggs of *M. incognita* were extracted with 20% bleach and collected on a 500-mesh sieve. The collected eggs were separated using 35% (*w*/*v*) sucrose by centrifugation at 4200× *g* for 5 min and rinsed three times with sterile water. Then, the eggs were surface sterilized with 10 % bleach for 5 min followed by rinsing with sterile water 4–5 times. The surface-sterilized eggs were hatched in an incubator at 25 °C for two days. The J2 stage larvae of *M. incognita* were collected, counted, and resuspended in sterile water for subsequent experiments.

1 mL of *P. oligandrum* culture filtrate and 150 J2s of *M. incognita* were added to a 24-well cell culture plate and incubated at 25 °C. Nematode viability was observed at 2 h, 4 h, 6 h, 8 h, 10 h, 12 h, and 24 h, and the number of dead J2s (determined based on a lack of mobility) were counted. Seven time points were chosen to facilitate a high temporal resolution of the mortality rate. The non-inoculated V8 medium was used as a control, with three replicates for each treatment.

### 2.3. Assays for Root Attraction of Nematodes and Nematode Penetration of Roots

The seeds of tomato cv. Moneymaker, which is susceptible to RKNs, were surface-sterilized with 10% bleach solution for 15 min, vigorously shaken and washed with sterilized water 5–6 times, and germinated in the dark on sterilized filter paper for 4–5 d until the tomato root length was about 1.5–1.8 cm.

A 23% Pluronic F-127 gel was prepared and stored at 4 °C according to Gardener and Jones [24]. A total of 20,000 J2s were prepared by centrifuging at 5000 RPM for 5 min to remove most of the water, and the J2s’ suspension was mixed with the Pluronic F-127 gel at 4 °C for a final concentration of 500 J2s/mL [25]. To a six-well cell culture plate, 3 mL of the Pluronic F-127 gel (containing 1500 J2s) was added dropwise to each well, so that the gel filled the bottom of the well, and excess air bubbles were carefully removed. Tomato seedling roots (root length about 1.5–1.8 cm) were immersed in Po_GAQ1 or Po_CBS530.74 culture filtrate, or non-inoculated V8 liquid medium, and shaken at 80 RPM for 30 min at 28 °C. The filtrate-treated tomato seedlings were placed into the Pluronic F-127 gel with two seedlings placed in each well. Three replicates were set up for each treatment, and the attraction of tomato roots to J2s was observed and imaged under a stereomicroscope at 0 h, 2 h, 4 h, 6 h, 8 h, 10 h, 12 h, 24 h, 36 h, 48 h, 60 h, and 72 h. The nematodes penetrating the tomato seedling roots were counted at 24 h using acid fuchsin staining. The acid fuchsin stain was prepared as a 0.35% *w*/*v* acid fuchsin solution in 25% *v*/*v* glacial acetic acid, and the destaining solution was prepared by mixing acetic acid, propanetriol, and water (1:1:1). For the acid fuchsin staining, the tomato seedling roots infected with nematodes, from the 24 h time point, were taken from the Pluronic F-127 gel and were treated with 20% bleach (Clorox, Oakland, CA, USA) for 3 min. The tomato seedling roots were gently rinsed with tap water for 1 min to remove most of the bleach, and then soaked in sterile water for 15 min to remove residual bleach. The 0.35% acid fuchsin solution was diluted to a 40 mL 0.00875% solution using sterilized water, and the roots were added and heated in a microwave until gently boiling. After waiting for them to cool to room temperature, the roots were removed, and 10 µL of the destaining solution was added for observation. The experiment was repeated three times.

### 2.4. Greenhouse Pot Trial of P. oligandrum Control of RKN Disease in Tomato

For the greenhouse pot trial experiment, surface-sterilized tomato (cv. Moneymaker) seeds were sown in sterilized soil (9:1 nutrient soil: vermiculite ratio) and germinated under 25 °C and 16 h light and 8 h dark conditions in a growth chamber, and, 2 weeks later, they were transplanted to sterilized soil in 0.56 L pots and grown in a greenhouse. The conditions in the greenhouse were set to 16 h of supplemental lighting and 8 h of darkness, and a temperature of 25 ± 5 °C and a humidity of approximately 70–80%. Preventative biocontrol treatments were carried out 3 d after the transplanting of tomato by inoculating the tomato rhizosphere with a 50 mL volume of Po_GAQ1 or Po_CBS530.74 mycelial pellets or a water control. A mycelium-based inoculum was used for *P. oligandrum* strains because oospore production was not optimized to achieve a yield that could be practically used as an inoculum. Seven days after applying the above treatments, each tomato seedling was inoculated with a suspension containing 500 *M. incognita* J2s at five points on its roots according to the “five-point method”, and the number of tomato root galls was counted at 30 d after inoculation with J2s of *M. incognita*. Note that the 7 d wait after the inoculation with *P. oligandrum* strains was considered sufficient for *P. oligandrum* to colonize the tomato rhizosphere. From plants inoculated with the biocontrol treatments, but not with RKN, measurements were taken to measure any growth promoting effects of the treatments. The plant height (from the base to the top of the stem), plant fresh weight, and root wet weight were measured. To measure root wet weights, the roots were first washed with tap water, and then excess water was removed with tissue paper. 

### 2.5. RNA Sequencing and Analysis

Square 10 cm Petri dishes containing 20 mL of 1% water agar were used to germinate ten tomato (cv. Moneymaker) seedlings. When seedlings reached a 1.5 cm root length, a 0.6 cm diameter mycelial plug of Po_GAQ1 was placed 0.5 cm from the tomato root tip, and incubated for 24 h at 25 °C. For the control, the seedlings were not inoculated with a mycelial plug. A 1 cm portion of the root tip was cut using a scalpel blade, and the seedling root tip samples were flash-frozen in liquid nitrogen and then stored at −80 °C. The experiment was performed with three replicates. RNA was extracted using a Trizol-based method (Accurate Biology, Changsha, China), and samples were stored at −80 °C. The samples were ground in 2 mL tubes with liquid nitrogen, and 1 mL of TRIzol reagent (Accurate Biology, Changsha, China) was added, and the sample was mixed and incubated for 5 min, and then 200 μL of chloroform was added, the sample was shaken, incubated for 5 min, and then centrifuged for 15 min at 12,000× *g* at 4 °C. The supernatant was transferred to a new tube, and an equal volume of pre-cooled isopropanol was added and mixed well, and then allowed to stand for 30 min at −80 °C, and then centrifuged at 12,000× *g* for 10 min at 4 °C. The pellet was washed with 1 mL of 75% ethanol, and later dissolved in 30 μL of DEPC-treated water. RNA samples were analyzed by Nanodrop OD260/280 and OD260/230 to check their quantity and purity. The sequencing libraries for transcriptome sequencing were made with 1.5 μg of total RNA using the NEBNext^®^ Ultra™ RNA Library Prep Kit from Illumina^®^ (NEB, Ipswich, MA, USA), following the manufacturer’s instructions. Poly-T oligo-attached magnetic beads were used to purify the mRNA, and the mRNA was fragmented with divalent cations and heat using the NEBNext First Strand Synthesis Reaction Buffer (NEB, Ipswich, MA, USA). Fragments of 200–250 bp were purified using the AMPure XP system (Beckman Coulter, Brea, CA, USA), and then 3 μL of the USER Enzyme (NEB, Ipswich, MA, USA) was used on size-selected, adaptor-ligated cDNA at 37 °C for 15 min, and then for 5 min at 95 °C, and then PCR was carried out with high-fidelity DNA polymerase along with the Universal PCR primers and Index (X) Primer. The Agilent Bioanalyzer 2100 system was used to assess the quality of the purified PCR products and library. The sequencing was carried out on an Illumina NovaSeq 6000 platform by Allwegene Technology (Beijing, China), and paired-end 150 bp reads were made. About ~40 M raw reads for each sample were made.

Clean data (clean reads) were obtained, using in-house Perl scripts, by removing reads containing adapter or poly-N and reads that were otherwise low-quality. The Q20, Q30, GC content, and sequence duplication level of the clean data were determined. These clean reads were then mapped to the reference genome sequence for the tomato cultivar ‘Heinz 1706’ genome version build SL4.0 using STAR aligner (v2.5.2b) [26]. Only reads with either a perfect match or one mismatch were analyzed further and annotated based on the reference genome for the tomato cultivar ‘Heinz 1706’ genome version build SL4.0 and annotations from the International Tomato Annotation Group (ITAG) version ITAG4.0 [27,28]. The percentage of uniquely mapped reads was at least 95% in all samples. HTSeq (v 0.5.4) [29] was used to count the number of reads mapped to each gene. Gene expression levels were estimated by fragments per kilobase of transcript per million fragments mapped (FPKM). Differential expression analysis was performed using DESeq2 (1.14.1) [30], and *P_adj_* < 0.05 from DESeq2 analysis was used to designate a gene as differentially expressed. The PCA was performed using FactoMineR [31] in the R statistical environment. GOSeq was used for gene ontology enrichment analysis with a corrected *p* value cut-off of ≤0.05 [32]. KOBAS was used for KEGG pathway enrichment with a corrected *p* value cut-off of ≤0.05 [33].

### 2.6. RNA Extraction, cDNA Synthesis, and Real-Time PCR Analysis

For qPCR analysis, RNA was extracted using a TRIzol-based method (Accurate Biology, Changsha, China), according to the manufacturer’s protocol and as described in the previous section on RNAseq. First-strand cDNA was synthesized from 1000 ng of total RNA using the Evo M-MLV RT Mix Kit with gDNA Clean for qPCR Ver.2 (Accurate Biology, Changsha, China Product no. AG11728), and the residual genomic DNA in the RNA template was removed at the same time. 

cDNA was used as the template for qPCR using a Roche LightCycler^®^96 (Roche Diagnostics, Rotkreuz, Switzerland) real-time fluorescence qPCR machine. The reaction mix was as follows: 10 μL of SYBR Green Premix Pro Taq HS qPCR Kit master mix (Accurate Biology, Changsha, China, Product no. AG11701), 0.4 μL of each primer (10 μM), 1 μL of cDNA, and 8.2 μL of nuclease-free water, for a total volume of 20 μL. The qPCR cycling conditions were as follows: pre-denaturation at 95 °C for 180 s, 40 cycles of 95 °C for 10 s, 60 °C for 30 s, followed by melt curve analysis. The specificity of the PCR was determined by the melt curve analysis and by sequencing the qPCR products, and the relative expression of the genes was calculated using the 2^−ΔΔCt^ method [34]. Actin, ubiquitin, and 18S were used as the internal reference genes, and expression levels were normalized using the arithmetic mean of the *Cq* values of the three reference genes. All primers used are listed in Appendix A, along with their primer amplification efficiencies, which were between 92% and 106%.

### 2.7. Statistical Analysis

Generally, experiments were repeated three times and subjected to ANOVA analysis using GraphPad Prism statistical software package (version 10.1.0, GraphPad Software, San Diego, CA, USA). Tukey’s least significant difference post hoc test or Dunnett’s post hoc test was used for multiple mean comparisons at the *p* ≤ 0.05 level.

## 3. Results

### 3.1. The P. oligandrum GAQ1 Culture Filtrate Is Lethal to M. incognita

Initially, to identify whether *P. oligandrum* could effectively control nematode-caused diseases, the culture filtrate was tested for its inhibition of *M. incognita* second-stage juveniles (J2s). The culture filtrate of Po_GAQ1 led to an 84.17% mortality of J2s 24 h after treatment. Another *P. oligandrum* strain, Po_CBS530.74, had a J2 mortality rate of 74% 24 h after treatment. The nematode J2 mortality rate of the control group was only 14.5% 24 h after treatment with non-inoculated V8 liquid medium (Figure 1). As the initial experiments supported an inhibitory effect of the *P. oligandrum* culture filtrates towards *M. incognita*, the next experiments tested whether this inhibitory effect could protect tomato seedlings from colonization and penetration by *M. incognita*.

### 3.2. P. oligandrum Culture Filtrate Inhibits J2s’ Movement toward Tomato Roots

1500 second-stage juveniles (J2s) of *M. incognita* were dispersed in Pluronic F-127 gel, and the aggregation of J2s at the tomato root tips was observed at different time points (Figure 2). In the control group, at 8 h, the number of nematodes attracted to the roots gradually increased, and there was a tendency for them to aggregate (Figure 2). At 24 h, after the number of nematodes aggregated at the tip of the tomato root began to decline, root expansion at the tip was noticeable, indicating that many nematodes had entered into the tip of the root. The results of the assays on the root attraction of nematodes (chemotaxis) showed that, compared with the control, the Po_GAQ1 and Po_CBS530.74 culture filtrates inhibited the attraction ability of tomato to nematodes. 

At the 24 h time point, the tomato root staining (Figure 3) showed that Po_GAQ1 filtrate-treated tomato root tips had the lowest number of penetrating nematodes (23.47 ± 10.4), which was a reduction in penetration of 78.2% compared to the non-filtrate-treated control group, where an average of 107.74 ± 34.1 nematodes penetrated the root. For the *P. oligandrum* strain Po_CBS530.74 filtrate treatment, the average number of nematodes penetrating the root was 30.42 ± 13.9, which was a reduction of 71.76% compared to the control group.

### 3.3. P. oligandrum Reduces RKN Infestation and Promotes Plant Growth in Pot Trials

To investigate whether *P. oligandrum* could control the RKN diseases of tomato under greenhouse conditions, Po_GAQ1 and Po_CBS530.74 mycelial pellets were used to inoculate tomato plants in a pot experiment, and, a week later, the tomato plants were infected with *M. incognita*. The results showed that, compared with the control group, both Po_GAQ1 and Po_CBS530.74 significantly (*p* < 0.05) reduced the number of tomato root galls per gram of root by 57.87% and 46.28%, respectively (Figure 4). 

The treatment of tomato plants with Po_GAQ1 and Po_CBS530.74 also promoted the growth of the tomato plants (Figure 5). One week post treatment, the biomass of tomato plants inoculated with the *P. oligandrum* strains began to increase significantly, and the size of the increases compared to the water-inoculated control became larger over time. Four weeks after the treatment of the tomato plants with Po_GAQ1, there were significant increases in their measured plant height (41 cm, 36.4% increase), plant weight (40.8 g, 27.2% increase), and root weight (5.5 g, 48.2% increase) compared with the water-treated control. For the treatment with Po_CBS530.74, there were significant increases in the measured plant weight (42.2 g, 31.2% increase) and root weight (5.8 g, 56.6% increase) compared with the water-treated control, while its trend showed an increase in plant height (33.8 cm, 11.9% increase) that was not statistically significant in the post hoc test used. Overall, the treatments with Po_GAQ1 and Po_CBS530.74 led to clear growth-promotion effects on the tomato plants and demonstrated that there were plant-mediated effects of *P. oligandrum* and also that these growth-promotion effects could be another potential mechanism contributing to disease control.

### 3.4. Inoculation of Tomato Seedling Roots with P. oligandrum GAQ1 Leads to Gene Expression Changes in Tomato Roots

The results of the pot trial with tomato and Po_GAQ1 showed a disease control effect, and while the nematicidal activity of the filtrates supported a direct effect of *P. oligandrum* on nematode viability, the increase in the growth of tomato plants inoculated with *P. oligandrum* also suggested that there was an interaction between *P. oligandrum* and tomato plants. To investigate the response of tomato to *P. oligandrum*, the gene expression changes in tomato seedling roots growing with the Po_GAQ1 strain were measured. 

The PCA showed a clear separation of the three replicate transcriptome samples from the control roots and the samples from roots growing in the presence of *P. oligandrum* (Figure 6A). The clustering pattern of the replicates in the PCA showed a clear effect of the presence of *P. oligandrum* on the gene expression patterns in the tomato seedling roots. In total, there were 2719 differentially expressed tomato genes, with 1437 genes up-regulated in the presence of *P. oligandrum* and 1282 genes down-regulated in the presence of *P. oligandrum* (Figure 6B and Appendix A). GO enrichment and KEGG pathway analyses were used to understand the gene expression changes in tomato roots due to the presence of *P. oligandrum*.

There were 191 GO terms enriched in the up-regulated genes and 27 GO terms enriched in the down-regulated genes (Appendix A). Of the GO terms enriched in the up-regulated genes, three of the most notable terms were the GO:0006952 defense response, the GO:0050832 defense response to fungus, and the GO:0098542 defense response to other organisms. These three GO terms supported a response from defense-related tomato genes to the presence of *P. oligandrum*. The GO terms for chitinase activity (GO:0004568) and the salicylic acid biosynthetic process (GO:0009697) were also enriched in the tomato genes up-regulated when inoculated with *P. oligandrum*. There were 11 genes responsible for the enrichment of the GO term for chitinase activity, including seven putative GH19 chitinases (Pfam00182) and three putative GH18 chitinases (Pfam00704) (Appendix A). Among the notable GO terms enriched in the down-regulated genes, the lignin (GO:0046274) and phenylpropanoid (GO:0046271) catabolic processes’ GO terms were enriched in the down-regulated tomato genes when tomato seedling roots were inoculated with *P. oligandrum*.

In the up-regulated genes, there were three KEGG pathways that were enriched: the sly03010 ribosome pathway, sly00520 amino sugar and nucleotide sugar metabolism pathway, and sly00100 steroid biosynthesis pathway (Appendix A and Appendix A). Other notable KEGG pathways that, while significantly enriched at the non-corrected P-value level, were not significant after correction for multiple testing were the sly00940 phenylpropanoid biosynthesis pathway, sly04016 MAPK signaling pathway—plant, and sly04626 plant–pathogen interaction pathway (Appendix A and Appendix A). No KEGG pathways were significantly enriched in the down-regulated genes when using the corrected *p* value (*q* value) threshold.

### 3.5. Presence of P. oligandrum GAQ1 Modulates the Expression of Tomato Defense, Growth, and Signaling-Related Genes

Based on the trends from the GO and KEGG pathway enrichment analyses, a subset of tomato genes that were differentially expressed were selected for further analysis of their expression changes from a time course of tomato seedlings inoculated with Po_GAQ1. Appendix A shows representative images of the tomato seedling roots sampled for the time-course analysis. Auxins are key plant hormones for mediating growth, and PIN family proteins are auxin efflux transporters and are involved in controlling cellular auxin efflux [35]. The auxin efflux facilitator *SlPIN10* gene (*Solyc04g056620.2*) was up-regulated at all the time points analyzed (12 h, 24 h, 36 h, and 48 h) (Figure 7). The expression or abundance of pathogenesis-related proteins, such as Pathogenesis-related protein P2 (PRP2) and Pathogenesis-related protein 1, can change due to the presence of pathogens or pathogen-derived molecules. The tomato genes pathogenesis-related protein P2 (PRP2) (*Solyc01g097240.3*) and pathogenesis-related protein 1 (*Solyc00g174340.2*) were up-regulated at multiple time points in the tomato seedling roots inoculated with *P. oligandrum* (Figure 7). *WRKY* genes are key regulators of plant biotic and abiotic responses [36,37]. The tomato *WRKY33* gene (*Solyc09g014990.3*) was up-regulated at all of the time points analyzed (12 h to 44 h). Jasmonate ZIM-domain (JAZ) proteins are important regulators of jasmonate signaling, which is an important component of the plant responses to pathogen attacks [38]. The jasmonate ZIM-domain protein 1 (*Solyc12g009220.2*) was up-regulated at multiple time points in seedling roots inoculated with *P. oligandrum* (Figure 7).

The genes used in this time-course analysis were also used to validate the expression changes in the RNAseq data using the same samples that were used for the RNAseq, and here, as expected, the trends from the RNAseq data were validated by the qPCR analysis (Appendix A).

## 4. Discussion

Here we have shown that *P. oligandrum* has a biocontrol effect on RKN-caused diseases of tomato, and that the nematicidal activity (leading to the reduced attraction of nematodes to tomato roots and their penetration of tomato roots), the promotion of tomato plant growth, and the priming of tomato plant defenses, are all likely contributory mechanisms. To our knowledge, this is the first study of a biocontrol oomycete species controlling an RKN-caused disease of tomato. 

There are previous reports of biocontrol bacterial and fungal species producing metabolites such as enzymes, toxins, antibiotics, and volatiles with nematicidal activity to control root-knot nematode diseases; for recent reviews see [17,18]. For example, volatile organic compounds produced by the fungus *Metarhizium brunneum* can kill juveniles of the RKN *M. hapla* [39] and the *Bacillus subtilis* strain Bs-1 can produce nematicidal compounds to repel *M. incognita* J2s [40]. By studying the effect of *P. oligandrum* culture filtrates on the activity of *M. incognita* J2s, it was found that *P. oligandrum* culture filtrates had a good nematicidal effect (Figure 1), and their specific nematicidal components could be investigated in future studies. For example, the *Pseudomonas simiae* MB751 strain caused the mortality of *M. incognita* J2s, and a cyclic dipeptide Cyclo(L-Pro-L-Leu) was found to be one of the main nematicidal components produced by the strain [41]. Their nematicidal components likely include hydrolytic enzymes such as proteases (including collagenases) and chitinases [42]. The genome of the *P. oligandrum* ATCC 38472 strain was annotated with 156 secreted proteases [43], and these could potentially contribute to the nematicidal properties of the *P. oligandrum* GAQ1 and CBS530.74 culture filtrates by hydrolyzing the *M. incognita* cuticle. Also of note, the genome of the *P. oligandrum* CBS530.74 strain was annotated with 13 GH19 enzymes which are predicted to have chitinase activity [44]. Although this chitinolytic activity is less relevant to its nematicidal activity towards *M. incognita* J2s, it could contribute to its nematicidal activity towards *M. incognita* eggshells.

Plant root secretion is an important nutrient for rhizosphere microorganisms and plays a signal-mediated role in the recognition process between rhizosphere microorganisms and hosts. Studies have shown that plant root secretions influence nematodes’ growth and development, movement orientation, and host infestation (see the review by Rutter, Franco and Gleason [11]). Biocontrol agents have the potential to adversely affect plant-parasitic nematodes’ growth and development by influencing the composition of root secretions. We found that tomato seedling roots treated with a culture filtrate of *P. oligandrum* could significantly reduce the attractiveness of their roots to *M. incognita*, and this likely led to the reduction in nematode penetration (Figure 2 and Figure 3). The relative contribution of direct nematicidal activity from the *P. oligandrum* filtrates, compared to their indirect effects via modulating root secretions to reduce the attraction of roots to *M. incognita,* was not possible to determine in our study but would be of interest for future studies. 

There are broad similarities in the ~50% reductions in the numbers of galls in tomato using *P. oligandrum* strains (Figure 4) compared with other studies from the literature where fungal or bacterial biocontrol agents were used. When a *T. harzianum* strain was used to treat tomato plants before an inoculation of the RKN *M. incognita*, there was a 74% reduction in the number of galls per gram of root [20]. The treatment of tomato with *Pseudomonas simiae* MB75 before its inoculation with *M. incognita* led to a ~5-fold reduction in its galling index [41]. Similarly, in a study using another bacterial biocontrol strain, the treatment of tomato with *Bacillus velezensis* strain YS-AT-DS1 before its inoculation with *M. incognita* led to a ~50% reduction in the number of galls per plant [45]. In a screen of potential bacterial biocontrol agents, the treatment of tomato with two of the best-performing strains, *Bacillus methylotrophicus* strain R2-2 and *Lysobacter antibioticus* strain 13–6, before its inoculation with *M. incognita*, led to a ~50% reduction in their gall-number-based disease index [46]. Our greenhouse pot-based experiments showed that the number of tomato root galls decreased significantly after seedlings’ treatment with cultures of either of the *P. oligandrum* strains, supporting their potential to control RKN infestations in field conditions.

Through the transcriptomics and analysis of their differentially expressed genes, some plant-mediated mechanisms of the *P. oligandrum* GAQ1 suppression of root-knot nematode disease can be suggested. One of the first studies that demonstrated that *P. oligandrum* could induce plant defense responses was a study with tomato [47]. Several of the tomato transcriptomic responses to *P. oligandrum* support a tomato defense-priming response to the presence of *P. oligandrum*. Lignin can function as a barrier to pathogen entry, and lignin accumulation can increase the resistance of *A. thaliana* to *M. incognita* penetration [48]. The transcriptome responses of tomato seedling roots suggested attempts to increase lignification in their roots. Several genes in the phenylpropanoid pathway were up-regulated, which could increase the supply of sub-units for the synthesis of lignin, and several laccase genes were also down-regulated, which could be involved in reducing the breakdown of lignin. Two early studies on *P. oligandrum* suggested that a deposition of phenolic compounds in tomato cell walls was one of the responses to *P. oligandrum* [47,49]. Salicylic acid is a key hormone in plant defense responses [50]. In the tomato seedling roots inoculated with *P. oligandrum*, there was an up-regulation of genes involved in the synthesis of salicylic acid (Figure 6). Previously, in rapeseed seedlings which were inoculated with various *P. oligandrum* strains before sowing, higher levels of salicylic acid were measured [51]. Chitinases are one of the major categories of pathogenesis-related (PR) genes with representatives in the PR families PR-3, PR-4, PR-8, and PR-11, and they are an important part of plant defenses against chitin-containing pathogens [52]. Eleven tomato putative chitinase genes were up-regulated in the seedling roots in response to *P. oligandrum* (Appendix A). These up-regulated chitinases could contribute to their increased resistance to *M. incognita* by degrading *M. incognita* eggshells, as chitin is an important component of nematode eggshells [42]. Other tomato PR genes from the family PR1 and the PRP2 gene were up-regulated, suggesting a broad spectrum of the up-regulation of PR genes in response to *P oligandrum* and possibly contributing to priming tomato defenses. Recently, in tomato, a PR1 family protein called SlPR1 has been suggested to be important for early immunity against Fusarium wilt disease [53]. It is likely that the tomato’s defense priming in part occurs via the constitutively expressed *P. oligandrum* elicitin-like protein oligandrin, as oligandrin has been shown previously to induce defense responses when applied to tomato fruit [54], and also possibly via *P. oligandrum* Nep1-like proteins (NLPs) which were shown recently to reduce the *P. capsici* infection of tomato fruits [55]. Previously, volatiles from the *P. oligandrum* GAQ1 strain were shown to up-regulate genes involved in growth-related hormone signaling and stress responses in ginger [56,57]. Note that the contribution of *P. oligandrum*-produced volatiles to the tomato seedling roots’ gene expression is probably limited as the volatile compounds likely diffuse out from the culture dishes.

Part of our motivation for investigating *P. oligandrum* was that the oomycete species had not been previously tested on root-knot nematodes. However, there are reports on using *P. oligandrum* biocontrol strains to control other nematode-caused diseases. Previously, a commercial preparation of *P. oligandrum* called Ecosin was shown to have larvicidal activity towards the nematode *Uncinaria stenocephala,* which parasitizes animals [58]. The *P. oligandrum* preparation Ecosin can also control the roundworm infection in dogs and cats caused by the ascarid nematodes *Toxocara canis* and *T. cati* [8]. The mechanisms by which Ecosin works here with these nematodes appear to involve nematophagous activities towards eggs and larvae after the germination of *P. oligandrum* from the spores contained in the Ecosin preparation [8].

In conclusion, we have demonstrated how a biocontrol species from the phylum Oomycota, *P. oligandrum*, can also function in antagonizing root-knot nematodes with similar levels of efficacy to fungal and bacterial biocontrol agents in controlling nematode-caused diseases or infestations of tomato.

## Figures and Tables

**Figure 1 jof-10-00265-f001:**
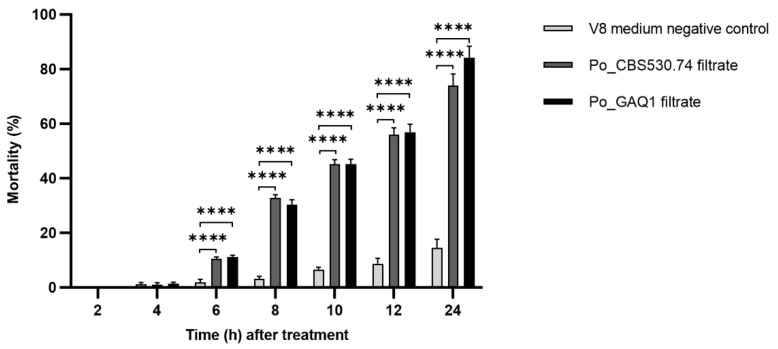
Nematicidal effects of culture filtrates of *P. oligandrum* (Po) strains Po_CBS530.74 and Po_GAQ1 towards *M. incognita* J2s at various time points after the addition of the filtrates to J2s. An non-inoculated V8 medium was used as a control. Error bars represent standard deviation, *n* = 4. **** = *p* < 0.0001. The results of repeated experiments all showed the same trend.

**Figure 2 jof-10-00265-f002:**
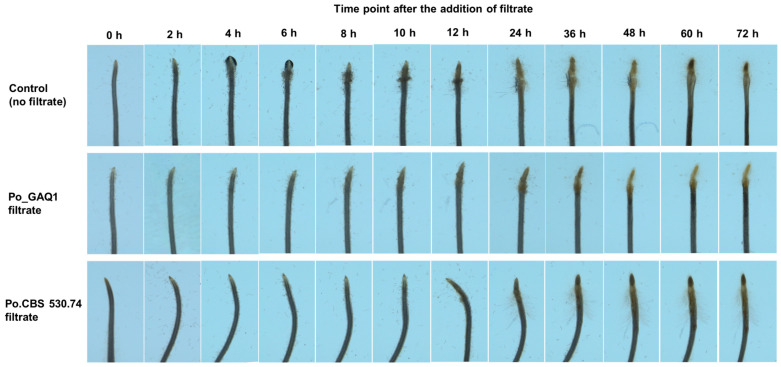
Effect of culture filtrates of *P. oligandrum* (Po) strains Po_CBS530.74 and Po_GAQ1 on the attraction of *M. incognita* J2s to tomato roots. The controls are the seedling roots in Pluronic F-127 gel without a culture filtrate treatment. The images are representative of the seedling roots that were analyzed for each treatment at each time point. The results of repeated experiments all showed the same trend.

**Figure 3 jof-10-00265-f003:**
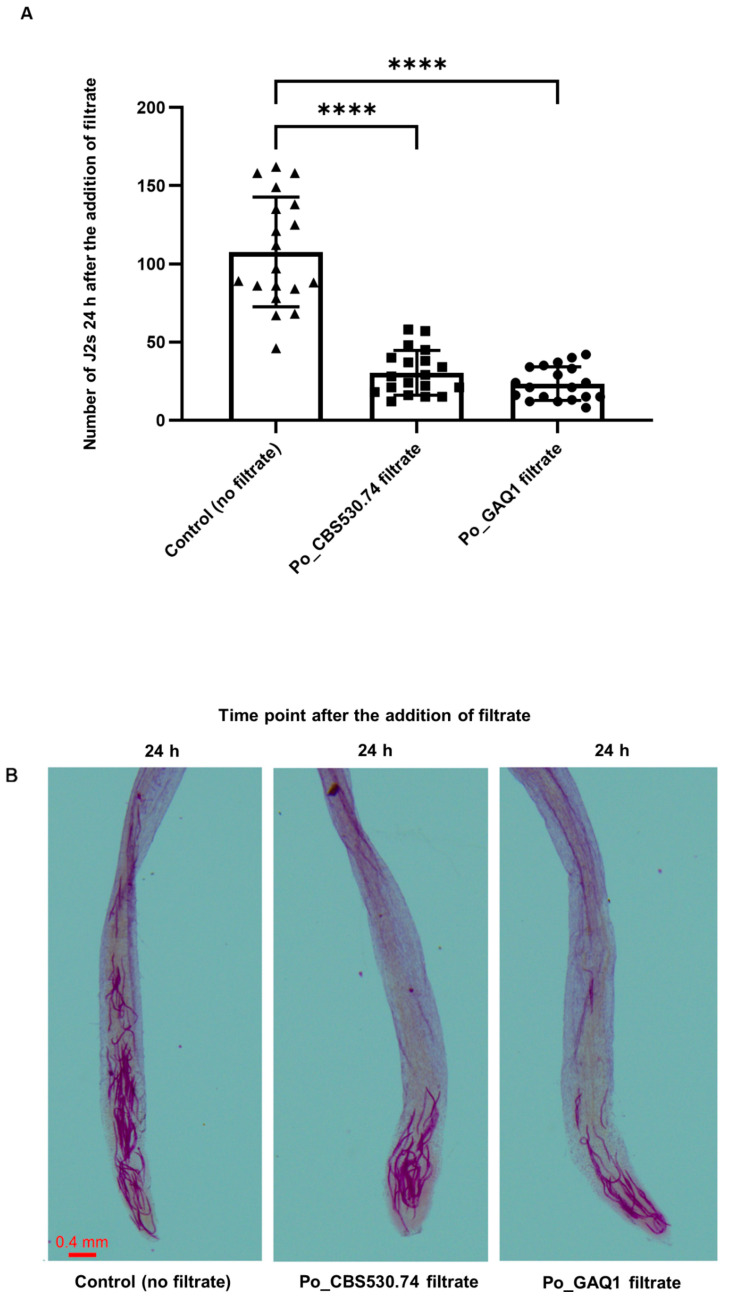
Infection or penetration of tomato roots by *M. incognita* 24 h after the addition of culture filtrates of *P. oligandrum* (Po) strains Po_CBS530.74 and Po_GAQ1. (**A**) number of J2s of *M. incognita* infecting a tomato root at 24 h and (**B**) representative images of acid fuchsin staining of nematodes in tomato roots. Error bars represent standard error, *n* = 19 tomato seedling roots. Significant differences, from ANOVA analysis, are shown: **** = *p* < 0.0001. The control is the seedling root in Pluronic gel without a *P. oligandrum* culture filtrate treatment. The results of repeated experiments all showed the same trend.

**Figure 4 jof-10-00265-f004:**
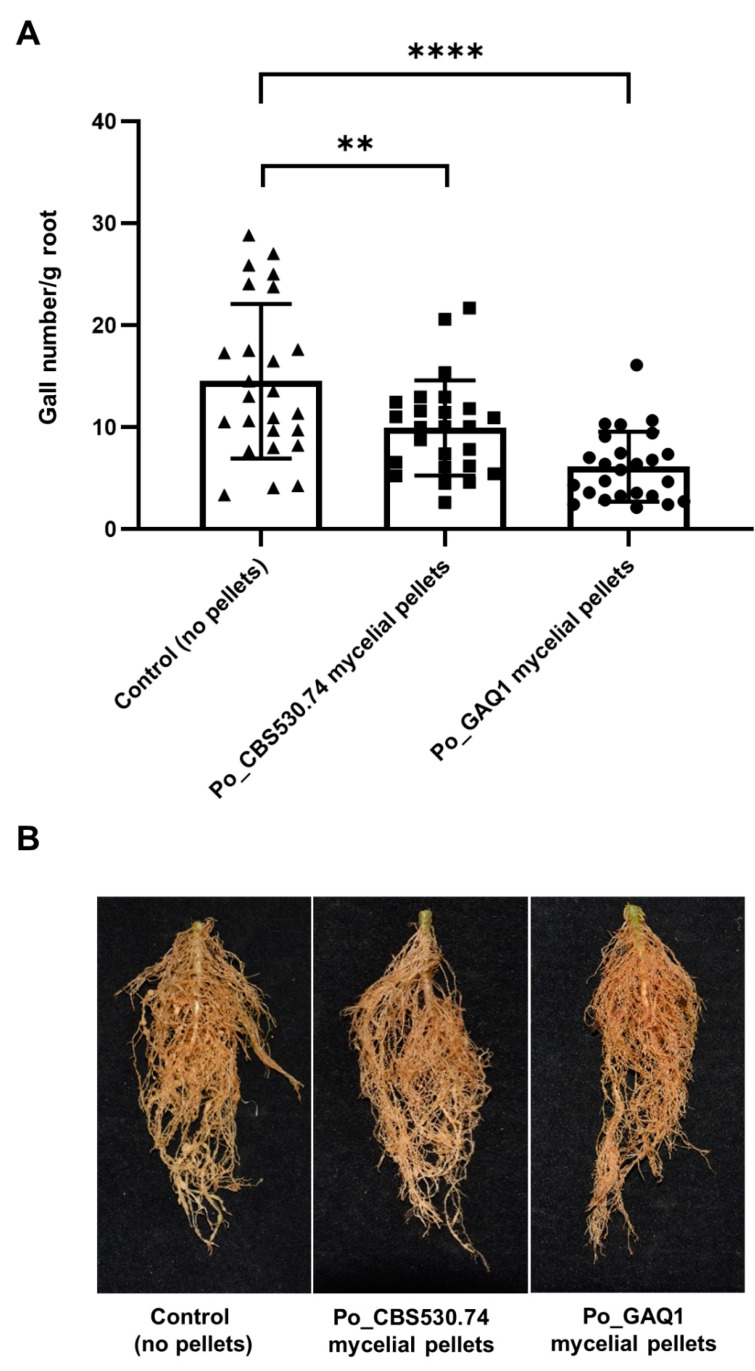
Biocontrol effect of mycelial pellets of *P. oligandrum* (Po) strains Po_CBS530.74 and Po_GAQ1 on root-knot-nematode-caused tomato disease in pot trial. (**A**) the number of galls per gram of roots, and (**B**) representative images of plants from pot trial. Error bars represent standard deviation, *n* = 25. Significant differences, from ANOVA analysis, are shown: **** = *p* < 0.0001, ** = *p* < 0.01. The control here is infected tomato plants without a *P. oligandrum* treatment. The results of repeated experiments all showed the same trend.

**Figure 5 jof-10-00265-f005:**
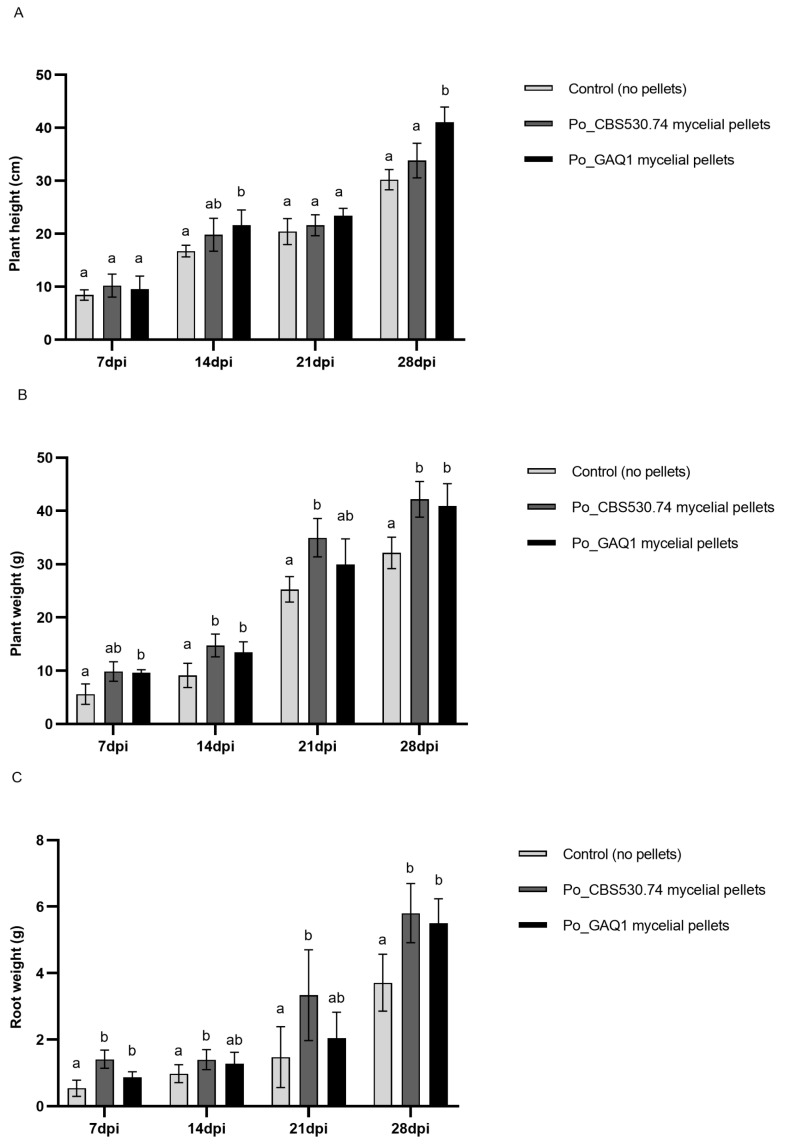
Growth promotion effect of mycelial pellets of *P. oligandrum* (Po) strains Po_CBS530.74 and Po_GAQ1 on tomato. (**A**) Plant height and (**B**) plant and (**C**) root fresh weights were measured. The data are mean ± SE (*n* = 5). Different letters indicate significant differences among different treatments at a time point according to a one-way ANOVA (*p* < 0.05). The control here is tomato plants without a *P. oligandrum* treatment. Note that the plants in this growth promotion trial were not inoculated with *M. incognita*.

**Figure 6 jof-10-00265-f006:**
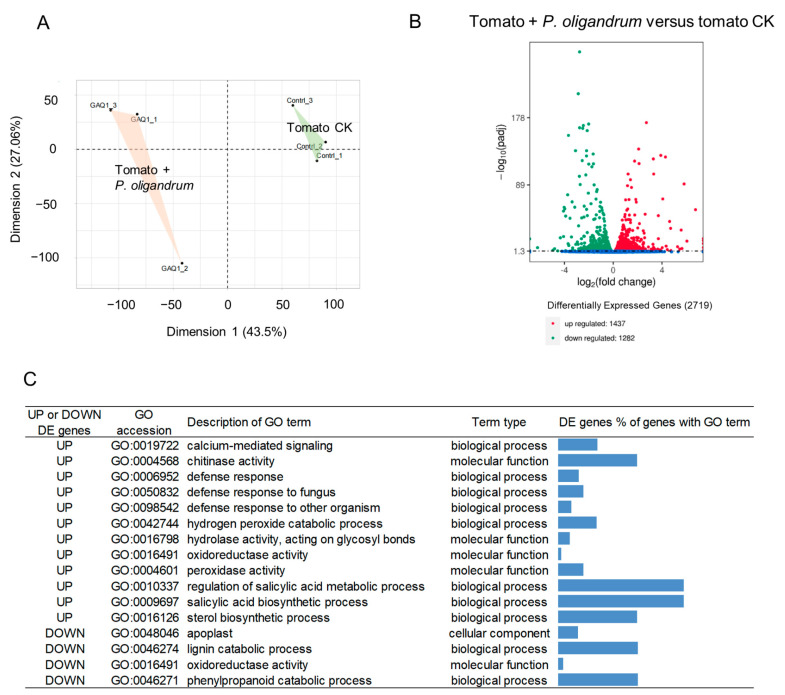
Overview of RNAseq data of tomato seedlings’ roots with and without Po_GAQ1. (**A**) Principal component analysis of the transcriptome samples, (**B**) volcano plot of the up-regulated and down-regulated genes, and (**C**) list of notable enriched GO terms in the up- and down-regulated genes.

**Figure 7 jof-10-00265-f007:**
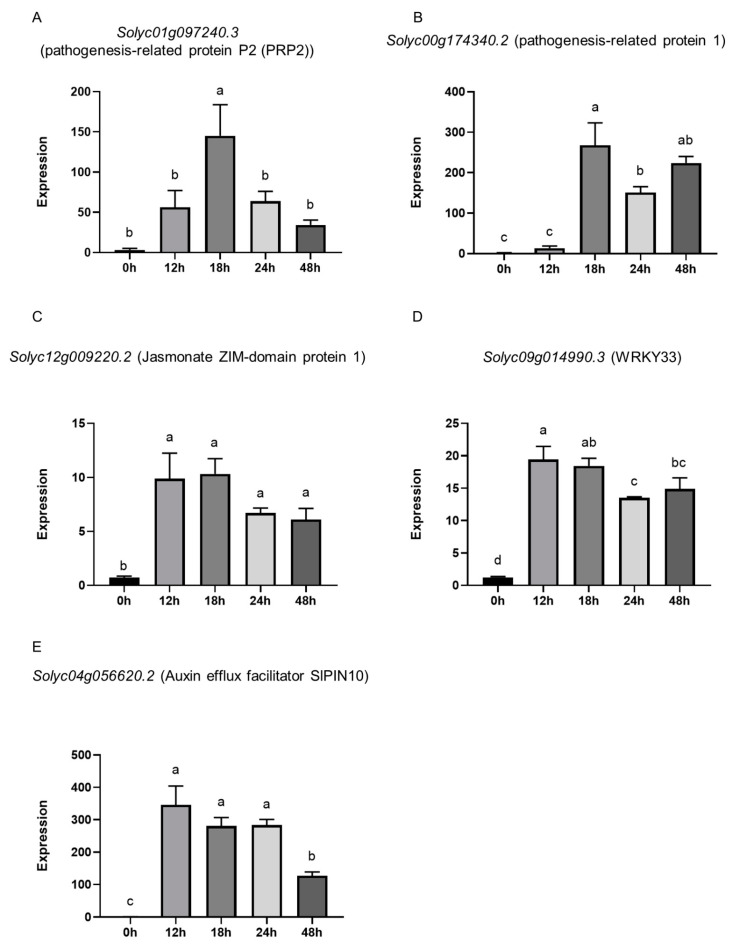
Time course of gene expression in tomato roots with Po_GAQ1 compared to roots without Po_GAQ1. The expression of the genes (**A**) *Solyc01g097240.3*, (**B**), *Solyc00g174340.2*, (**C**) *Solyc12g009220.2*, (**D**) *Solyc09g014990.3*, and (**E**) *Solyc04g056620.2* was measured. Each bar represents their expression in tomato roots growing with Po_GAQ1 at the indicated time point compared to tomato roots growing without Po_GAQ1 at 0 h. For the qPCR analysis, three reference genes (actin, ubiquitin, and 18S) were used as the internal reference genes. Different letters indicate significant differences among different treatments according to one-way ANOVA (*p* < 0.05) followed by Dunnett’s post hoc test to compare 0 h to the other time points for seedlings inoculated with *P. oligandrum*. Error bars represent the standard error (*n* = 3).

## Data Availability

The RNAseq data from this study was deposited in the NCBI GEO database with the accession number GSE262653.

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
