# Peer review of "Biological Control of a Root-Knot Nematode Meloidogyne incognita Infection of Tomato (Solanum lycopersicum L.) by the Oomycete Biocontrol Agent Pythium oligandrum"

_jof, 2024, doi:10.3390/jof10040265_

Round 1

Reviewer 1 Report

Comments and Suggestions for Authors

Dear authors, see inside of the document my suggestions/comments to be considered.

Reviewer 2 Report

Comments and Suggestions for Authors

A review report:

The biocontrol agent Pythium oligandrum, a member of the phylum Oomycota, has demonstrated efficacy in controlling diseases caused by a wide range of plant pathogens, including fungi, bacteria, and oomycetes. However, its potential to control diseases caused by plant root-knot nematodes (RKNs) remained unknown. This study focuses on investigating the effectiveness of two P. oligandrum strains, GAQ1 and CBS530.74, against RKN Meloidogyne incognita infections in tomato plants (Solanum lycopersicum L.).

P. oligandrum culture filtrates exhibited high lethality, causing up to 84% mortality in M. incognita J2s after 24 hours, compared to 14% in the control group. Tomato roots treated with P. oligandrum culture filtrates demonstrated reduced nematode attraction, with a concurrent decrease of up to 78% in the number of nematodes penetrating the roots. In the greenhouse pot trial, P. oligandrum GAQ1 inoculation significantly reduced gall formation by 58% in M. incognita-infected tomato plants. P. oligandrum GAQ1 mycelial treatment resulted in notable increases in tomato plant height (36%), weight (27%), and root weight (48%). Transcriptome analysis identified approximately 2,500 differentially expressed genes in tomato seedling roots inoculated with P. oligandrum GAQ1, indicating modulation of plant hormone-signaling and defense-related pathways.

The study supports the potential of P. oligandrum GAQ1 as a biocontrol agent against Meloidogyne incognita in tomato plants. Multiple mechanisms contribute to this biocontrol effect, including the direct inhibition of M. incognita, potential priming of tomato plant defenses, and the promotion of plant growth. These findings hold significance for sustainable agriculture practices and warrant further exploration of P. oligandrum's applications in nematode management.

Comments to authors

The Materials and Methods section in the manuscript appears comprehensive in detailing the procedures conducted for the study. However, there are a few shortcomings that could be addressed to enhance clarity:

Strain Descriptions and Source Clarity:

The information provided about the P. oligandrum strains (GAQ1 and CBS530.74) lacks certain key details, such as specific characteristics, genetic traits, or unique attributes that could aid in better understanding or replicating the study. The origin of strain GAQ1 is briefly mentioned as isolated from soil in China, but more details about the location, soil conditions, and any potential environmental factors influencing its isolation would enhance the section's completeness.

Culturing Conditions:

The description of routine plate culturing and preparation of culture filtrates is concise, but specific details on the composition of the 10 % V8 juice agar and liquid medium are missing. Including detailed recipes and specific conditions for agar and liquid medium preparation would improve the reproducibility of the experiments.

Nematode Viability Assessment:

The time points chosen for observing nematode viability after exposure to P. oligandrum culture filtrates (2 h, 4 h, 6 h, 8 h, 10 h, 12 h, and 24 h) appear frequent, possibly leading to redundant data. A justification or rationale for the chosen time intervals would be beneficial.

Root Attraction and Penetration Assays:

The description of seed germination, Pluronic F-127 gel preparation, and J2s suspension preparation is clear. However, details regarding the rationale behind choosing specific time points for observation (0 h, 2 h, 4 h, etc.) need clarification. Additionally, the methodology lacks information on how the results were quantified or assessed statistically. The acid fuchsin staining method for counting nematodes penetrating tomato seedling roots is described, but specific steps or details about the staining process are missing.

Greenhouse Pot-Trial Design:

The description of the greenhouse pot-trial lacks clarity regarding the specific growth conditions provided, such as light duration, temperature, and humidity. These factors are crucial for understanding the environmental context of the experiment.While preventative biocontrol treatments are briefly mentioned, the reasons behind choosing the specific application method, timing, and dosage require elucidation for better understanding and replication.

RNA Sequencing and Analysis:

The RNA sequencing section provides a detailed description, but certain aspects like the rationale for choosing the 'Heinz 1706' genome as a reference and specific details on the RNA extraction process could be expanded for clarity. The PCA analysis is mentioned but lacks information on the significance or implications of the observed results. A brief interpretation or connection to the study's objectives would enhance the section.

Real-Time PCR Analysis:

The qPCR analysis description is generally clear, but more details on primer design rationale, validation steps, and the criteria for choosing Actin as the internal reference gene are needed. The section would benefit from additional information on the statistical analysis method used, including the choice of ANOVA and Tukey's test, to aid readers in understanding the reliability of the obtained results.

Addressing these shortcomings would contribute to a more comprehensive and reproducible Materials and Methods section.

I recommend major revision!

Round 2

Reviewer 2 Report

The authors of the article have addressed all of my questions. I have no further comments and suggest that the article be accepted.

The authors of the article have addressed all of my questions. I have no further comments and suggest that the article be accepted.